# Motivations and willingness to provide care from a geographical distance, and the impact of distance care on caregivers' mental and physical health: a mixed-method systematic review protocol

Eva Bei ,[1] Mikołaj Zarzycki,[2] Val Morrison,[2] Noa Vilchinsky [1]

[1]Department of Psychology, Bar-Ilan University, Ramat Gan, Israel
[2]School of Psychology, Bangor University, Bangor, UK

**Correspondence to**
Eva Bei; eva.bei@biu.ac.il

## ABSTRACT

**Introduction** Distance caregivers (DCGs) are a growing population with substantial contribution to informal care. While a reasonable amount is known on the determinants of motives and willingness to provide local informal care, and the local caregiver outcomes, reports for the distance caregiving population are lacking. An evidence synthesis of what motivates and makes DCGs willing to care from a distance and the impact of that care on their mental and physical health would highlight any gaps or consensus in knowledge. This would guide the research needed towards the development of tailored interventions, in order to support DCGs and promote the sustainability of distance care.

**Methods and analysis** This protocol adheres to Preferred Items for Reporting of Systematic Reviews and Meta-Analyses Protocols guidelines and the Joanna Briggs Institute (JBI) Methodology for mixed-method reviews. A comprehensive search strategy will be conducted in four electronic databases (CINAHL, MEDLINE, PubMed and PsycINFO). Grey literature will also be assessed to minimise publication bias. Two independent reviewers will assess each study for inclusion and any discrepancies will be resolved with the consultation of a third reviewer. Eligible studies for inclusion will be English language studies exploring the motives and willingness to care for a care recipient with a chronic disease, disability or frailty from a geographical distance; or studies focusing on the mental and physical health outcomes of DCGs. Qualitative and quantitative data will be integrated in a single qualitative synthesis following the JBI convergent integrated approach. Study quality will be assessed using the Mixed Methods Appraisal Tool version 2018.

**Ethics and dissemination** Ethical approval is not required for this study as no primary data will be collected. Findings will be disseminated through peer-reviewed publication and presentations at academic conferences and lay summaries for various stakeholders.

**PROSPERO registration number** CRD42020156350.

## Strengths and limitations of this study

► The review will set the groundwork and inform research needed to guide the development of tailored interventions to support distance caregivers (DCGs) and promote the sustainability of distance care.

► The mixed-method design of the study will ensure a wide variety of data on distance care is captured from both qualitative and quantitative research findings.

► A rigorous, systematic approach will be applied to searching, screening, extracting and analysing evidence in four different academic databases and in grey literature.

► Anticipated potential limitations include high degree of heterogeneity across studies as there is no consensus on the definition of distance caregiving which makes it difficult to compare studies.

► The review will be restricted to studies published only in English, which may cause language bias.

## INTRODUCTION

With the ageing of the population worldwide, numbers of older and care-dependent adults are rising.[1–5] Informal caregivers, who provide unpaid care to a family member or friend with long-term care needs, are an essential pillar for sustaining healthcare systems worldwide and maintaining outpatient care.[6–8] Traditional patterns of informal care usually involved family or friends coresiding or living close to the care recipient.[9] However, as the demand for caregivers continues to grow, social changes such as urbanisation, gendered roles within society, increased labour market mobility and globalisation have affected the traditional ways of providing care.[9–11] Distance care has emerged in those cases where adult caregivers are not staying with their care recipient but making efforts for providing informal care from a geographical distance.[11–13]

It is estimated that 15%–20% of all informal care is provided by distance caregivers (DCGs).[12] One of the problems in this field of research is that there is no consensus on

how to best define distance caregiving. Most research to date has used mileage[12] or travel time categories[14 15] to measure the geographical distance from the care recipient and define DCGs. Overall, travel time and space along with a number of socioeconomic factors which are confounded with distance, such as travel costs and access to transportation, will influence distance caregiving—in its existence as well as in its extent.[16]

In their previous review, Cagle and Munn[17] suggested that the definition of distance care as articulated by Parker *et al*[18] is the most comprehensive and compelling. Indeed, the definition appears to be appropriate as it operationalises geographical distance without focusing exclusively on distance and travel time but also on a number of other socioeconomic factors that determine geographical distance and affect distance care provision. According to Parker *et al*[18], a DCG could be defined as: 'Anyone (1) who provides informal, unpaid care to a person experiencing some degree of physical, mental, emotional, or economic impairment that limits independence and necessitates assistance; and (2) who experiences caregiving complications because of geographic distances from the recipient, as determined by distance, travel time, travel cost, personal mobility problems, limited transportation and other related factors that affect the caregiver's access to the care recipient'.

Demographical analyses show that most DCGs are adult children providing assistance to a family member, usually an aged parent, parent in-law or step-parent with a chronic health condition, disability or other long-lasting healthcare needs.[17 19] DCGs engage in many activities to ensure that the needs of their care recipient are met, such as providing social and emotional support, managing financial affairs, monitoring and coordinating care and healthcare services.[12 20–22] Although most DCGs are unable to provide hands-on care on a daily basis as local caregivers, they are significantly involved in performing practical and nursing tasks when visiting their loved one. In fact, approximately 75% provide assistance with instrumental activities of daily living such as household tasks, cooking, grocery shopping, medical care and transportation.[12] Also, 40% report assisting with activities of daily living such as bathing, dressing, feeding and toileting.[12]

### Motivations and willingness to provide informal care

In recent years, a growing body of literature has explored caregivers' motivations and willingness to carry out a caregiving role.[23–25] Research on local caregiving has identified multiple and often inter-related determinants of caregivers' motives and willingness to care for a loved one.[23] Informal care is likely to be provided out of love and affection, reflecting a long-lasting family or friend relationship between the caregiver and the care recipient.[26 27] Reciprocity is also a frequently cited motive for informal care provision. Adult children or children in-law often reciprocate the care and love received in the past from the care recipient, by undertaking the caregiving role.[28–32] In addition, caregiving is frequently described as

a family duty, filial obligation or responsibility. Adult children are often motivated to care as a means of fulfilling their care responsibilities, in socio-cultural contexts where caregiving is viewed as an unspoken family value, a moral obligation.[28 33–37] Beside these factors, availability of other caregivers[30 34] and resources for formal care[38] may also have an effect on the provision of informal care.

Baldassar[21] demonstrated that DCGs may describe some similar motives to care to those of local caregivers. Despite the unique challenges and complications caused by physical separation, DCGs felt that they were greatly indebted to their parents for raising them and it was now their turn to reciprocate the care received and pay them back even from a distance.[21] They also reported a strong sense of obligation to care based on moral and societal expectations regarding caretaking duties.[21] However, DCGs may also report motives to care unique to the distance caregiving situation, for example, in a recent study,[39] DCGs expressed feelings of guilt about not being physically present to fulfil their duty to care. Although physical distance limited DCGs from providing care on a regular basis, it also worked as a strong motivator, increasing their willingness and efforts for providing adequate care from afar.[39]

Previous findings have highlighted the significance of motivations and willingness to take on the caregiving role. Camden and colleagues[40] found that those who were unwilling to provide care, reported higher abusive behaviours towards the care recipient while their loved one was more likely to be admitted to a care home the following year. Another qualitative study showed that caregivers who provided care out of societal expectations had less control over the caregiving challenges.[41] Additionally, Romero-Moreno *et al*[24] reported that extrinsic motives for caring (ie, filial obligation) are associated with poorer caregivers' mental health. Similarly, in their systematic review, Quinn *et al*[25] found that caregivers motives to provide informal care can have implications on their psychological well-being.

### Mental and physical health outcomes of informal caregiving

Studies of broader caregiving samples suggest that caregivers can experience both positive and negative outcomes as a result of their caretaking responsibilities.[42–44] The positive aspects of caregiving may include an enhanced relationship with the care recipient, feelings of personal growth, gains and satisfaction and an overall rewarding and meaningful experience.[8 42 45–47] However, caregivers may also experience negative outcomes related to the caregiving role, such as poor mental and physical health.[47 48] Numerous studies have identified high levels of caregiver burden across a wide range of chronic diseases, as a multidimensional response to the various stressors associated with caring.[48–50] In terms of psychological morbidity, high levels of emotional distress and increased symptoms of anxiety and depression have been reported[51–53] with, for example, a meta-analytical review reporting a prevalence of anxiety and depressive

symptomatology among caregivers of stroke survivors as above 20% and 40%, respectively.[52] In addition, many caregivers experience poor physical health, related to their role.[54–56] A meta-analysis of 84 studies conducted across a range of health conditions revealed that caregivers reported statistically significant lower levels of well-being and physical health than non-caregivers,[55] with some specific findings pointing to caregivers showing poorer immune response[57] and exhibiting greater cardiovascular reactivity[56] than non-caregivers.

Geographical distance creates additional burdens for what is already often a stressful care work.[17] Koerin and Harrigan[58] found that almost 80% of DCGs reported emotional distress related to feelings of inadequacy on how to assess the needs of their loved one from afar and uncertainty regarding the progression of their illness. Additionally, in their recent study, Li et al[14] found that adult child primary caregivers who live more than 30 min away from their frail older parents had higher levels of depression than coresiding caregivers. Findings also showed that although coresiding caregivers were more distressed by objective caregiver burden, DCGs were more distressed by subjective caregiver burden which includes perceptions and attitudes related to caregiving.[14]

The negative mental and physical health outcomes of caregiving can threaten caregivers' commitment to the welfare of the care recipient and reduce the quality of care provision.[59 60] Additionally, poor caregiver outcomes are associated with adverse health outcomes for care recipients, including higher levels of emotional distress, elder abuse, mortality and hospitalisation.[61–63] Increased morbidity of caregivers can also lead to high individual and societal burdens and affect the sustainability of informal care.[63 64]

### Aim of the review

DCGs are a growing population who make a substantial contribution to informal care. To date, to the best of our knowledge, no review has systematically evaluated, and quality appraised the current evidence on motivations and willingness to care from a distance and the impact of distance care on caregivers' mental and physical health outcomes. Most systematic reviews have currently focused on local caregivers,[23 24 65] excluding the distance caregiving population. In addition, past reviews of distance caregiving have been limited to evaluating evidence on the phenomenon of distance care in general, focusing on the definition and sociodemographic characteristics of DCGs and describing only briefly the benefits and costs of such a role.[17 66] Other reviews have exclusively focused on distance caregiving for patients with advanced cancer without exploring distance care on other health conditions with different care needs or demands[13]; or on the availability of technology-based and eHealth interventions to support DCGs.[67]

However, as previously stated, DCGs may report unique experiences and motives to care, associated with the geographical and spatial distance from the care recipient.[21 39] Experiencing the added challenges and complications of caring from afar, DCGs may also be at higher risk of poor mental and physical health outcomes.[14 17 58] A systematic review would synthesise evidence from studies on what motivates DCGs and makes them willing to provide care from a distance, and what is the impact of that care on their mental and physical health. Further understanding of these issues, would provide evidence for the development of geographically sensible and tailored interventions in order to promote the sustainability of distance care and support those caring from afar. Therefore, this review aims to:

1. Synthesise and critique the evidence on the determinants of motivations and willingness to care from a geographical distance.
2. Synthesise and critique the evidence on the impact of distance caregiving on caregivers' mental and physical health.

## METHODS AND ANALYSIS

This protocol is guided by the Preferred Items for Reporting of Systematic Reviews and Meta-Analyses Protocols (PRISMA) checklist[68] (online supplemental appendix 1) and the Joanna Briggs Institute (JBI) methodology for mixed-methods systematic reviews.[69]

### Study registration

Based on the PRISMA guidelines,[68] the protocol for this systematic review was registered on the international database of prospectively registered systematic reviews in health and social care, PROSPERO. Any important protocol amendments will be recorded in PROSPERO and published with the results of the review.

### Eligibility criteria

#### Types of studies

Eligible studies for inclusion will be English language peer-reviewed studies or unpublished studies such as doctoral theses in order to reduce publication bias. These will include quantitative studies (analytical observational and descriptive observational studies), studies that focus on qualitative data and mixed-method study designs. The inclusion of different types of research will provide more informative findings for each of the two review objectives and increase the ability of those findings to inform future research, policy and practice.

### Participants

The population of interest are adult DCGs (aged 18 and above) of adult family members or friends with healthcare needs. The comprehensive definition of distance care by Parker et al[18] was used and adapted for the purposes of this review, and in order to maximise our access to empirical studies that conceptualised geographical distance from the care recipient in different ways. Thus, as a DCG could be defined:

1. Anyone who provides informal, unpaid care to a relative or friend with a chronic illness, disability or frailty that limits independence and necessitates assistance.

2. Who experiences caregiving complications because of geographical distances from the care recipient, as determined by distance, travel time, travel cost, personal mobility problems, limited transportation or other related factors that affect the caregiver's access to the care recipient.

Care recipient's health conditions eligible for this review will include any chronic illness and disability (eg, cancer, dementia, cardiovascular disease, stroke) or simply frailty. Studies exclusively focusing on young DCGs (under 18 years old) or DCGs of children and adolescents will be excluded, as young caregiving[70] and care receiving[71] experiences are associated with additional needs and burdens.

## Outcomes
### Primary outcomes

The primary outcomes of interest are caregivers' motives and willingness to provide care from a geographical distance. These may include qualitative and/or quantitative data. Qualitative findings describing various determinants of motives and willingness to care from a distance such as for example family values, love and affection, reciprocity, guilty feelings, filial obligation and specific illness characteristics will be reported. In addition, qualitative studies exploring willingness to perform a variety of caregiving tasks from afar or when visiting the care recipient, such as emotional, nursing and instrumental care tasks, will be considered. For quantitative findings, studies must report on motives and willingness to care from a distance, assessed using a validated self-report instrument (eg, Motivations in Elder Care Scale,[72] Willingness to Care Scale[73]) or a self-report instrument developed for the purposes of the included study. Given that many studies explored the determinants of motives and willingness to provide local care using qualitative methodology, it is envisaged that our outcomes will be also primarily explored through qualitative data.

### Secondary outcomes

Secondary outcomes of interest are those studies assessing the impact of distance care provision on caregivers' mental and physical health outcomes, specifically depression, anxiety, emotional distress, caregiver burden, perceived physical health or physical symptoms. Eligible studies for inclusion will be quantitative studies that assessed distance caregiver outcomes using a validated self-report outcome instrument or an instrument developed for the purposes of the included study. Examples of validated psychrometric instruments for the secondary outcomes of interest include the Centre for Epidemiologic Studies-Depression Scale,[74] the Hospital Anxiety and Depression Scale,[75] the Profile of Mood States,[76] the Perceived Stress Scale,[77] the Zarit Burden Interview[78] or the Cohen-Hoberman Inventory of Physical Symptoms.[79] Qualitative findings exploring manifestations and perceptions of emotional distress, anxiety, depression, burden and physical health will be also considered.

## Search strategy

A detailed search strategy without limits on publication date, will be conducted on four relevant electronic databases (CINAHL, MEDLINE, PubMed and PsycINFO), to comprehensively search for studies on distance caregiving. Grey literature, including the OpenGrey (http://www.opengrey.eu/) and Electronic Theses Online Service (https://ethos.bl.uk/) databases, will be also searched to maximise our access to potentially relevant studies and reduce publication bias. In addition, the reference lists of all studies included will be screened to identify additional citations of interest.

The search strategy was developed using key terms related to the distance caregiving population and the primary and secondary outcomes of interest. The search was also informed by past reviews on distance care.[13 17] A variation of controlled vocabulary, relevant Medical Subject Headings when possible and free-text terms contained in the title/abstracts of publications will be applied. An example of the search strategy string terms used for PubMed database is presented (online supplemental appendix 2). Search strategy string will be then adapted to the remaining three electronic databases. Per database these terms will be mapped to subject headings. Included studies will be restricted to those written in English for ease of interpretation.

## Study selection

Following the search, all identified citations will be imported into Zotero and duplicates removed. In the first stage of primary screening, one reviewer (EB) will screen the titles of each study based on the inclusion criteria and when these are deemed relevant, she will go through abstracts. In addition, a second author (NV) will be consulted on the primary screening process using the eligibility criteria, and any doubts or conflicts for the titles or abstracts that are deemed relevant will be resolved through discussion. Once narrowed down by abstracts, a full-text review process will be completed in duplicate by two reviewers (EB and MZ) for studies that met the eligibility criteria at screening and for studies with unclear relevance. A third independent author (NV) will be consulted to resolve any discrepancies that arose between the two reviewers during the full-text process. Additional records identified from searches of grey literature and reference lists will also be assessed in detail against the eligibility criteria by the principal researcher (EB). Original authors of studies identified will be contacted if the full-text paper was not available or the relevance of a paper was unclear. The PRISMA flow diagram will outline the process of study selection (figure 1).[68] Reasons for exclusion of full-text studies that do not meet the eligibility criteria will also be recorded in the flow diagram.

## Assessment of methodological quality

To assess the methodological quality of the included studies, the Mixed Methods Appraisal Tool (MMAT) version 2018 will be used, a quality appraisal instrument

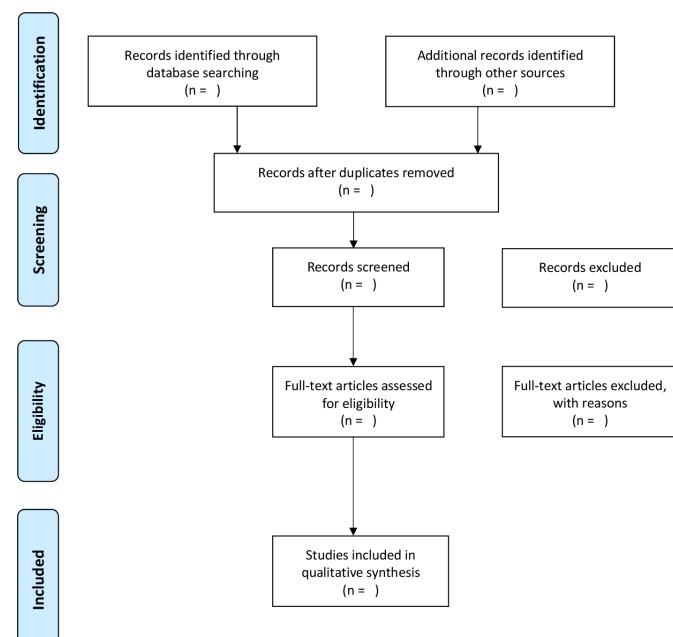

**Figure 1** Flow diagram of the planned study selection process adapted from the Preferred Reporting Items for Systematic Reviews and Meta-Analyses statement.[68]

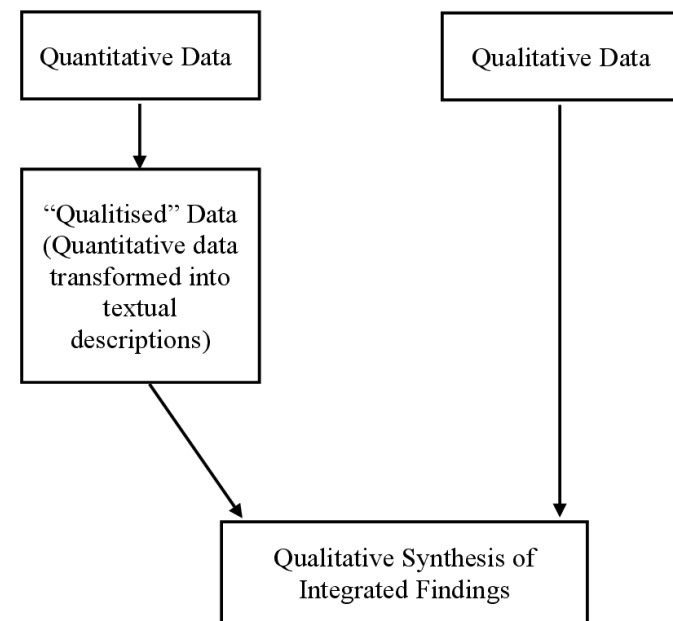

**Figure 2** Qualitative synthesis of integrated 'qualitised' quantitative data and qualitative data, following the convergent integrated approach as proposed by Joanna Briggs Institute methodology for mixed-method reviews.[69]

for mixed-method reviews.[80] The MMAT has been content validated and piloted across all methodologies.[80] The instrument consists of two screening questions for all study designs and five sections of specific questions regarding study type. The quality of each study will be assessed as low (0–1/5 criteria met), medium (2–3/5 criteria met) or high (4–5/5 criteria met), using the total number of MMAT criteria that were met. Quality appraisal will be completed in duplicate by two independent reviewers (EB and MZ). Any discrepancies will be resolved with the consultation of a third reviewer (VM). The strength of the body of evidence will be discussed and narratively incorporated into the synthesis. No study will be excluded on the basis of their methodological quality, as they could still offer valuable insight. Yet, findings of the studies assessed as low will be interpreted with caution.

### Data extraction
Data extraction process will be carried out using the JBI data extraction tool[69] as adapted and modified for the purposes of this review (online supplemental appendix 3). Data will be extracted for each study by one reviewer (EB) and independently extracted by the second reviewer (MZ) for accuracy, employing a double coding on portion of 80% of the included studies. Any discrepancies will be resolved with the consultation of a third reviewer (NV). Data extracted from the included studies will comprise research aims and objectives, study methods, population characteristics, phenomena of interest and context-related information, main study findings and relevant outcomes. Extracted information will vary across different study designs.

### Data synthesis
Data synthesis will involve a convergent integrated approach, as per JBI methodology for questions that can be addressed by both quantitative and qualitative research designs (figure 2).[69] Extracted data from quantitative findings will be first converted into 'qualitised data'. This will involve a narrative interpretation of the quantitative results and the transformation of numerical data into textual descriptions. According to the JBI guidelines, 'qualitising' quantitative data are recommended, as codifying quantitative data is less error-prone than attributing numerical values to extracted data of qualitative studies. At the simplest level, 'qualitised' data will comprise the description of sample based on descriptive statistics such as average or percentage scores. For quantitative data with a temporal or longitudinal component and those that explore associations using inferential statistics, 'qualitising process' will involve the identification of the variables included in the data analysis using textual descriptions and numerical data transformation. The 'qualitised' data will then be assembled and pooled with the results of qualitative studies. Similar to the meta-aggregative approach for JBI qualitative reviews, a detailed examination of the pooled data will be finally undertaken to identify categories based on similarity and produce a set of integrated findings for each of the two review objectives.[69]

### Patient and public involvement
Patients and members of the public were not involved in the design and development of this protocol.

## Ethics and dissemination

Ethical approval and consent to participate are not required for the proposed systematic review as no primary data will be collected. Findings will be disseminated through publication in an international peer-reviewed journal and presented at local and international scientific conferences on informal care and health psychology. A lay summary of the review will be written for healthcare organisations and non-scientific audiences such as caregivers and their care recipients and disseminated through mass emails, social media, blogs and appropriate webpages.

**Contributors** EB contributed to the conception and design of the study and wrote the manuscript. NV contributed to the conception and design of the study and critically revised the manuscript draft. MZ and VM critically revised the study design and the manuscript draft. All authors approved the final manuscript.

**Funding** This work is funded by the European Union's Horizon 2020 research and innovation programme under the Marie-Sklodowska Curie grant agreement no 814072.

**Competing interests** None declared.

**Patient consent for publication** Not required.

**Provenance and peer review** Not commissioned; externally peer reviewed.

**ORCID iDs**
Eva Bei http://orcid.org/0000-0002-3093-0829
Noa Vilchinsky http://orcid.org/0000-0003-4965-4745

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
