## [Reviewer comments · BMJ Open]

ARTICLE DETAILS

TITLE (PROVISIONAL)	Motivations and willingness to provide care from a geographic distance, and the impact of distance care on caregivers' mental and physical health: A mixed-method systematic review protocol
AUTHORS	Bei, Eva; Zarzycki, Mikołaj; Morrison, Valerie; Vilchinsky, Noa

VERSION 1 – REVIEW

REVIEWER	Peter Lucassen Radboud University Medical Center, Department of Primary and Community Care
REVIEW RETURNED	11-Nov-2020

GENERAL COMMENTS	Thank you for the opportunity to review this manuscript. It concerns a systematic review of qualitative and quantitative studies about distance caregivers' motives and willingness to provide long distance care and about the health consequences of providing distant care. Qualitative and quantitative results will be synthesized. The ultimate aim is to improve conditions for Distance Caregivers. In my opinion this is an important study. The authors have clearly and completely described the relevance of the study, the search strategy, the participants and the outcomes. I have some questions: Study selection: why did the authors decide to do the first selection based on title and abstract by (only) one reviewer. If I understand the text well, the second reviewer is involved after the first selection. Why not involve both reviewers in the first selection? Assessment of methodological quality: why have the authors chosen to assess methodological quality with the MMAT tool, and not, for example with one of the tools specifically designed for qualitative research such as SRQR or COREQ (see Equator Network). The MMAT tool is a fairly simple tool that has to be used for quality assessments of (probably) a range of different qualitative studies such as individual interview studies, focus group studies, ethnographic studies and so on. How could such an instrument be appropriate for this wide range of methodologies? An additional question is about the expertise of the research group. It is important to know how experienced (members of) this group is in assessing quality of qualitative research (and also in analyzing and synthesizing) the qualitative studies. Data synthesis: the manuscript contains a short description of Data synthesis. However, this is a difficult part of the research. Noblit and Hare have described their well-known strategy for meta-ethnography and have provided much more information for each of their seven steps in the process. In my opinion the authors have provide the reader with much more detail about this very important part of the project, for which a very experienced team has to be involved.
---

	Strengths and limitations section: I would relocate this section to the end of the manuscript. And I would like to see more detail in the limitations, especially concerning the analysis of the qualitative data. Conclusion: important study of which the description should be improved on several points.
--	---

VERSION 1 – AUTHOR RESPONSE

Reviewer’s Comments to Authors:

Thank you for the opportunity to review this manuscript. It concerns a systematic review of qualitative and quantitative studies about distance caregivers’ motives and willingness to provide long distance care and about the health consequences of providing distant care. Qualitative and quantitative results will be synthesized. The ultimate aim is to improve conditions for Distance Caregivers. In my opinion this is an important study. The authors have clearly and completely described the relevance of the study, the search strategy, the participants and the outcomes. I have some questions:

Response: We appreciate your willingness to revise our manuscript with interest and thank you for all your comments and valuable suggestions for its improvement.

1. Study selection: why did the authors decide to do the first selection based on title and abstract by (only) one reviewer. If I understand the text well, the second reviewer is involved after the first selection. Why not involve both reviewers in the first selection?

Response: Thank you for this comment. Single screening of the titles and abstracts will be applied following standard practices of the study selection process (Gough, Oliver & Thomas, 2017). However, a second author (NV) will be consulted on the primary screening process using the eligibility criteria, and any doubts or conflicts for the titles or abstracts selected or deselected will be resolved through discussion. Once narrowed down by abstracts, a full-text review process will be completed in duplicate by the two reviewers (EB and MZ) for studies that met the eligibility criteria at the primary screening and for studies with unclear relevance. We have now elaborated on this in the revised manuscript, under “Study Selection”.

See now on page 10 Line 23-28:

“Following the search, all identified citations will be imported into Zotero and duplicates removed. In the first stage of primary screening, one reviewer (EB) will screen the titles of each study based on the inclusion criteria and when these are deemed relevant, she will go through abstracts. In addition, a second author (NV) will be consulted on the primary screening process using the eligibility criteria, and any doubts or conflicts for the titles or abstracts that are deemed relevant will be resolved through discussion.”

Gough D, Oliver S, Thomas J, eds. An introduction to systematic reviews. London: Sage 2017: 28.

2. Assessment of methodological quality: why have the authors chosen to assess methodological quality with the MMAT tool, and not, for example with one of the tools specifically designed for qualitative research such as SRQR or COREQ (see Equator Network). The MMAT tool is a fairly simple tool that has to be used for quality assessments of (probably) a range of different qualitative studies such as individual interview studies, focus group studies, ethnographic studies and so on. How could such an instrument be appropriate for this wide range of methodologies? An additional question is about the expertise of the research group. It is important to know how experienced

(members of) this group is in assessing quality of qualitative research (and also in analyzing and synthesizing) the qualitative studies.

Response: Thank you for this feedback.

We propose to employ the MMAT tool since it offers a simple, yet detailed critical appraisal tool that systematically assesses the methodological quality of included studies that can be of different study designs. It is our considered opinion that MMAT fully meets the purposes of our mixed-method review that includes quantitative, qualitative and mixed-method studies.

More specifically:

1) The MMAT tool is a critical appraisal tool designed for the appraisal stage of mixed-method systematic reviews (as with that we propose). It includes five core quality criteria for each of the following five categories of study designs: 1) qualitative, 2) randomized controlled, 2) nonrandomized, 4) quantitative descriptive, and 5) mixed methods.

2) The MMAT tool has been content validated and piloted across all methodologies and, importantly, it also includes specific criteria for mixed method studies, which is not often found in other risk of bias tools (Hong et al., 2013).

3) Regarding the assessment of qualitative studies, the MMAT uses the following methodological quality criteria: 1) Is the qualitative approach appropriate to answer the research question? 2) Are the qualitative data collection methods adequate to address the research question? 3) Are the findings adequately derived from the data? 4) Is the interpretation of results sufficiently substantiated by data? 5) Is there coherence between qualitative data sources, collection, analysis and interpretation? In Part II of the tool's user guide, indicators are also added for each of the quality criteria, including those for assessing qualitative study designs, in order to appropriately distinguish between the different methodologies in qualitative research (e.g. ethnography, phenomenology, focus groups etc) and minimise errors during the 'risk of bias assessment'.

Finally, the review team has significant experience as senior health and clinical psychology research academics in systematic review methodology, including mixed-method reviews, and the alternatives of scoping reviews and realist reviews. The supervisory authors have previously conducted and published systematic reviews using validated quality assessment tools and applying rigorous methodologies for synthesising data, based on the eligible study designs for inclusion, i.e. thematic synthesis for qualitative reviews (Holmes, Hughes & Morrison, 2014; Magklara, Burton & Morrison, 2014; Roberts et al., 2017; Vilchinsky, Ginzburg, Fait & Foa, 2017). In addition, the first author has undergone training to acquire all the necessary skills and knowledge for conducting a systematic review.

Holmes EA, Hughes DA, Morrison V. Predicting adherence to medications using health psychology theories: a systematic review of 20 years of empirical research. *Value in Health*. 2014;1:863-76.

Hong QN, Pluye P, Fàbregues S, et al. Mixed Methods Appraisal Tool (MMAT), version 2018. Canadian Intellectual Property Office, Industry Canada 2018:1-12.

Magklara E, Burton CR, Morrison V. Does self-efficacy influence recovery and well-being in osteoarthritis patients undergoing joint replacement? A systematic review. *Clinical rehabilitation* 2014;28:835-46.

Roberts JL, Din NU, Williams M, Hawkes CA, Charles JM, Hoare Z, Morrison V, Alexander S, Lemmey A, Sackley C, Logan P. Development of an evidence-based complex intervention for community rehabilitation of patients with hip fracture using realist review, survey and focus groups. *BMJ open*. 2017;1:e014362.

Vilchinsky N, Ginzburg K, Fait K, Foa EB. Cardiac-disease-induced PTSD (CDI-PTSD): a systematic review. *Clinical psychology review* 2017;1:92-106.

3. Data synthesis: the manuscript contains a short description of Data synthesis. However, this is a difficult part of the research. Noblit and Hare have described their well-known strategy for meta-ethnography and have provided much more information for each of their seven steps in the process. In my opinion the authors have provide the reader with much more detail about this very important part of the project, for which a very experienced team has to be involved.

Response: Thank you for this comment. While Noblit and Hare's meta-ethnography (1988) is one of the most influential methodologies for qualitative evidence synthesis in health and social care research, it does not fully meet the research objectives of our mixed-method review. The present review adopts a mixed-methodology approach to synthesis with the aim to include different types of research (not only eliciting qualitative data), in order to meet each of our two review objectives. Therefore, to synthesise appropriately the data arising from different study designs, our data synthesis will involve a convergent integrated approach, as per the Joanna Briggs Institute (JBI) methodology, a process of combining extracted data from quantitative and qualitative studies. We have tried to further clarify how data will be synthesised in the revised manuscript.

See now on page 12 Line 1-17:

"Data synthesis will involve a convergent integrated approach, as per JBI methodology for questions that can be addressed by both quantitative and qualitative research designs [69]. Extracted data from quantitative findings will be first converted into "qualitised data". This will involve a narrative interpretation of the quantitative results and the transformation of numerical data into textual descriptions. According to the JBI guidelines, "qualitising" quantitative data is recommended, as codifying quantitative data is less error-prone than attributing numerical values to extracted data of qualitative studies. At the simplest level, "qualitised" data will comprise the description of sample based on descriptive statistics such as average or percentage scores. For quantitative data with a temporal or longitudinal component and those that explore associations using inferential statistics, "qualitising process" will involve the identification of the variables included in the data analysis using textual descriptions and numerical data transformation. The "qualitised" data will be then assembled and pooled with the results of qualitative studies. Similar to the meta-aggregative approach for JBI qualitative reviews, a detailed examination of the pooled data will be finally undertaken to identify categories based on similarity and produce a set of integrated findings for each of the two review objectives."

Lizarondo L, Stern C, Carrier J, et al. Chapter 8: Mixed methods systematic reviews. In: Aromataris E, Munn Z, eds. Joanna Briggs Institute Reviewer's Manual. The Joanna Briggs Institute 2017.

4. Strengths and limitations section: I would relocate this section to the end of the manuscript. And I would like to see more detail in the limitations, especially concerning the analysis of the qualitative data.

Response: Thank you for this comment regarding relocation, however we were responding here to the BMJ Open guidelines for study protocols, where it is stated that the "Strengths and limitations of this study" section should be placed after the abstract. A limit of 5 points is also imposed.

With regards to the 2nd point, as mentioned above, we consider one of the strengths of this study to be the mixed-methodology approach adopted, followed by the proposed rigorous methodology of synthesising data from all study designs. The mixed-method design of the review, including data analysis and synthesis of different types of research, will ensure a wide variety of data on distance care is captured and that our interpretations will account for both qualitative and quantitative research findings. We have now elaborated on this in the revised manuscript, under "Strengths and limitations of this study".

See now on page 3 Line 5-6:

“The mixed-method design of the study will ensure a wide variety of data on distance care is captured from both qualitative and quantitative research findings.”

VERSION 2 – REVIEW

REVIEWER	Peter Lucassen Radboud University Medical Center, Department of Primary and Community Care
REVIEW RETURNED	19-Mar-2021
GENERAL COMMENTS	I thank the authors for their responses on my remarks. I have no further comments.